# Oral zinc sulphate reduces the recurrence rate and provides significant therapeutic effects for viral warts: A systematic review and meta-analysis of randomized controlled trials

Chen-Chi Wang[1], Wei-Xiang Wang[2], Po-Yuan Wu [ID][3,4¤]*

1 Department of Medical Education, China Medical University Hospital, Taichung, Taiwan, 2 School of Medicine, College of Medicine, Taipei Medical University, Taipei, Taiwan, 3 School of Medicine, College of Medicine, China Medical University, Taichung, Taiwan, 4 Department of Dermatology, China Medical University Hospital, Taichung, Taiwan

¤ Current address: Department of Dermatology, China Medical University Hospital, Taichung City, Taiwan
* wu.poyuan@gmail.com

## Abstract

Zinc plays a crucial role in maintaining immune balance in the human body. Zinc is believed to substantially affect cytokine synthesis and signaling; thus potentially combating viral infections, including Human Papillomavirus infection, through various mechanisms. Several randomized controlled trials (RCTs) have investigated whether oral zinc sulphate can improve viral warts; however, no comprehensive data is currently available. Therefore, we conducted a meta-analysis to evaluate the effectiveness of oral zinc sulphate for the treatment of viral warts. On July 1, 2024, we performed an extensive database search on PubMed, Embase, Web of Science, Cochrane CENTRAL, and ClinicalTrials.gov. Initially, 952 studies were identified, and after screening, 7 studies were included in the final analysis. Our findings showed that the total clearance rate of warts was significantly higher in the oral zinc sulphate group than in the control group (risk difference = 0.288; 95% CI = 0.087–0.489; p = 0.005). Subgroup analysis revealed that this therapeutic effect was more pronounced in individuals with low initial plasma zinc levels (risk difference = 0.767; 95% CI = 0.649–0.885; p < 0.001). Additionally, meta-regression showed that rise in zinc ion levels post-treatment were correlated with better treatment outcomes (coefficient = 0.0068; p < 0.001). Furthermore, for patients receiving traditional warts treatment, combining oral zinc sulphate significantly reduced the 6-month recurrence rate (log risk ratio = −1.043; 95% CI = −1.666 – −0.420; p = 0.001). The most common treatment-related side effects were nausea (risk difference = 0.562; 95% CI = 0.088–1.036; p = 0.020) and vomiting (risk difference = 0.205; 95% CI = 0.092–0.317; p < 0.001). Based on this evidence, oral zinc sulphate monotherapy offers notable benefits to those avoiding conventional treatments, and when combined with traditional therapies for viral warts, it can notably reduce recurrence rates over six months.

**Data availability statement:** All relevant data are within the manuscript and its Supporting Information files.

**Funding:** The author(s) received no specific funding for this work.

**Competing interests:** No authors have competing interests.

## Introduction

Nongenital warts can present as common warts, flat warts, and deep palmoplantar warts, and are common and benign diseases caused by Human Papillomavirus (HPV) infection, with a global prevalence of approximately 10%. Warts are twice as common in Caucasians as in Blacks or Asians, with the highest prevalence observed in school-aged children and young adults [1]. External genital warts (condyloma acuminatum) are also caused by HPV infection and are generally considered sexually transmitted diseases, with a prevalence of about 10–20%. [2]. Generally, in individuals with normal immune status, warts are self-limiting diseases; however, the time required for them to disappear varies greatly, ranging from a few months to several years. [3]. One cohort study found that two-thirds of warts resolved without treatment within 2-year period [4].

Zinc is an essential trace element in the human body and is involved in the composition and production of numerous enzymes and transcription factors, respectively. Zinc plays a crucial role in the immune system. Previous studies have shown that zinc deficiency in animals leads to lymphopenia, lymphocyte dysfunction, and thymic atrophy [5]. A previous study reported substantially lower plasma zinc concentrations in patients with verruca vulgaris than in healthy individuals [6]. Another study demonstrated that wart severity was greater in individuals with reduced plasma zinc levels than in those with normal concentrations [7].

Many treatments for warts are available, such as cryotherapy commonly used for nongenital warts, topical podophyllin and imiquimod for external anogenital warts, antiviral medications, and immunotherapies such as zinc supplements [8]. However, cryotherapy can cause pain, blistering, and scarring, while topical podophyllin or imiquimod may lead to local skin irritation and are not safe during pregnancy [9]. Therefore, simply taking seemingly harmless zinc supplements to treat viral warts appears to be an attractive option. Existing literature on oral zinc therapy for viral warts shows mixed findings: one clinical trial reported excellent therapeutic effects, with 86% of patients taking zinc supplements achieving complete wart clearance, while none were cured in the placebo group [10]. Other studies suggest mild to moderate benefits [11,12], while some oppose these findings, stating that zinc supplements neither cure viral warts nor provide additional benefits [13–15]. Since the conclusions of these studies are inconsistent and comprehensive research has been lacking in recent years, we conducted a systematic review and meta-analysis of the efficacy and additional benefits of oral zinc sulphate administration in treating viral warts.

## Materials and methods

### General guidelines

This meta-analysis was conducted in strict adherence to the latest PRISMA guidelines from 2020 (S1 Table) and has been registered on INPASY, with registration number INPLASY202480037. Ethical approval and written informed consent from the participants were not required because of the nature and methodology of the study.

## Eligibility criteria

The population (P), intervention (I), comparison (C), and outcome (O) for this study were as follows: P, human participants with skin or external genital warts; I, either taking oral zinc sulphate alone or a combination of oral zinc sulphate with traditional treatments for viral wart; C, either taking a placebo or using traditional treatments, e.g., topical drugs or cryotherapy, not limited to oral medications, for viral wart; and O, number of patients who achieved viral wart clearance and relapse after completing treatment and treatment-related adverse events.

The inclusion criteria for this study were as follows: 1) randomized controlled trials (RCTs) with human participants; 2) RCTs including the number of individuals who experienced viral wart clearance after treatment in both experimental and control groups; 3) placebo-controlled RCTs (without any age or treatment duration restrictions); 4) RCTs including those administered a dosage of zinc sulphate; and 5) studies on the oral application of zinc ions.

The exclusion criteria were as follows: 1) RCTs involving non-oral zinc sulphate supplementation; and 2) overlapping participants.

## Information source, search strategy, and selection process

Two authors independently conducted data searches using electronic databases. Databases used include PubMed, Embase, Web of Science, Cochrane CENTRAL, and ClinicalTrials.gov, and the following keywords were used: [("wart*" OR "viral wart*" OR "verruca vulgaris" OR "condyloma acuminatum" OR "verruca plana") AND ("zinc" OR "zinc sulphate" OR "zinc gluconate")]. The literature search included articles from the earliest available records up to July 1, 2024, without year and language restrictions. Keywords and search results in different databases was listed in "S2 Table". Additionally, to ensure that all the studies that met the inclusion criteria were included, we manually reviewed the references of the systematic reviews [16] and included them according to the inclusion criteria. If the two authors had different opinions during the search process, a third author was consulted to resolve any disagreements.

## Data collection process

Two authors independently extracted data from the screened studies, including demographic information, study design, and intervention methods for both the oral zinc sulphate and control groups, as well as the results of each study. In case of data from different time points after the treatment, we used the final experimental results for our analysis. Data extraction and merging of results from the various study arms were performed according to the instructions in the related chapter of the Cochrane Handbook for Systematic Reviews of Interventions [17,18]. The primary outcome of this study was the total number of patients who achieved clearance of viral warts after treatment. For one study [11] that only provided data on 75% clearance of viral warts, we considered the number of individuals with 75% or more clearance as those who achieved complete disappearance of viral warts. The secondary outcome was defined as the rate of relapse during the follow-up period as well as any treatment-related side effects.

## Study risk of bias assessment

In this study, we used the Cochrane risk-of-bias tool for randomized trials (version 2, RoB 2, London, United Kingdom) [19] to evaluate the methodology of the included studies. The items analyzed included the randomization process, adherence to interventions, missing outcome data, outcome measurement, selective reporting, and overall risk of bias. Regarding the standards for reviewing randomization, the allocation sequence must first be kept concealed; for example, the allocation process should be conducted by an external institution or organization. Additionally, the allocation sequence should be unpredictable, with the article mentioning methods such as simple randomization, blocked randomization, or any elements involving random assignment (e.g., computer-generated random numbers or dice rolling). Finally, baseline imbalances need to be assessed.

## Synthesis methods

Owing to the heterogeneity among the target groups in the included studies, we applied a random-effects model and used Comprehensive Meta-Analysis software (Biostat, Englewood, NJ, USA) for analysis. Statistical significance was defined as a two-tailed p value < 0.05. Risk difference and its corresponding 95% confidence intervals were used to evaluate the primary outcome. Secondary outcomes, including relapse and treatment-related side effects, are presented as log risk ratios and risk differences with their associated 95% confidence intervals. To assess the heterogeneity of the RCTs included in this study, we calculated the $I^2$ value and Cochran's $Q$ statistic. We categorized the degree of heterogeneity as low, moderate, or high, corresponding to $I^2$ values of 25, 50, and 75%, respectively [20]. Additionally, we conducted a subgroup analysis based on the patients' initial plasma zinc ion concentrations and whether traditional treatments for viral warts were combined. Meta-regression analyses focused on establishing a correlation between the magnitude of the increase in zinc ion levels after treatment and treatment effectiveness, as well as the relationship between disease duration and treatment effectiveness. In the subgroup and meta-regression analyses, treatment efficacy was expressed as a risk difference. To ensure the robustness of this meta-analysis, sensitivity analyses were performed using the "one-study removal method" to assess whether there was a statistically significant change in the summary effect size when each study was sequentially excluded [18]. The assessment of potential publication bias was conducted according to the guidelines provided in the Cochrane Handbook for Systematic Reviews of Interventions [21]. Funnel plots were constructed and examined using Egger's regression test.

## Results

### Study selection and characteristics

The PRISMA flowchart published in 2020 was used for study selection. Initially, we identified 952 studies. We then removed duplicates (n = 298), studies on unrelated topics (n = 554), and non-RCT studies (n = 60), leaving 40 studies. Next, we excluded studies without full-text availability (n = 2), those not administering zinc ions orally (n = 26), and studies with inappropriate experimental or control group designs (n = 5). This resulted in a final selection of 7 studies for analysis. The literature searching process is presented in "Fig 1", and all non-duplicate studies (n = 654) identified during the search, along with the reasons for their exclusion, are documented in "S3 Table".

The seven included RCTs comprised 594 participants, 73.6% of whom were female. Based on the available data, the average age was 31.64 ± 12.49 (standard deviation, SD) years. The study duration was mostly eight months [10,13,15], with the shortest study lasting one month [11]. Among the analyzed articles, four were from Iran and the remaining three articles came from Iraq, Egypt, and Mexico. Among these 7 studies, 4 compared the efficacy of oral zinc sulphate to placebo capsule. One study compared the combination of zinc sulphate and cryotherapy with cryotherapy alone [13]. Another study compared the effectiveness of zinc sulphate combined with cryotherapy, imiquimod, or podophyllin to that of cryotherapy, imiquimod, or podophyllin alone [15]. The other study compared the effects of oral zinc sulphate with those of no treatment [11]. Other study details, including initial plasma zinc ion concentrations, the dosages of zinc sulphate, treatment and follow-up durations, are documented in "Table 1".

### Risk of bias in studies

Regarding the overall risk of bias for the seven included RCTs, 42.9% were assessed as having a low risk of bias, 57.1% were rated as having some risk of bias, and none (0%) were classified as having a high risk of bias (Fig 2). Among the evaluated items, three studies were classified as having some risk of bias in the randomization process because they did not describe the concealment of random allocation. For outcome measurement, two studies were rated as having some risk of bias owing to the potential influence of pre-existing knowledge on result interpretation. The detailed scores for each item are recorded in "Table 2".

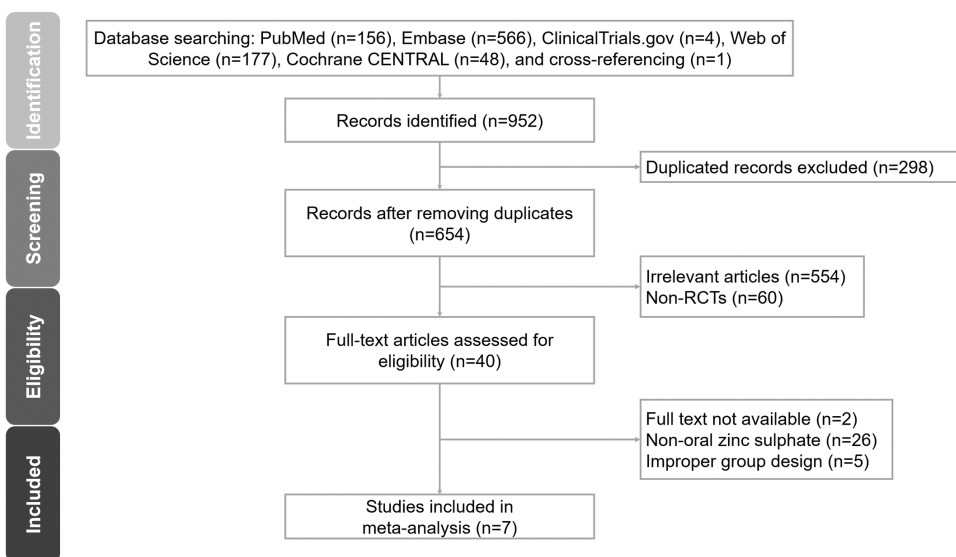

**Fig 1. PRISMA 2020 flowchart for the current meta-analysis.**

## Primary outcome

**The efficacy of oral zinc sulphate in treating viral warts.** Among the seven RCTs, the number of patients achieving total clearance of viral warts in the experimental groups, including those receiving oral zinc sulphate alone and zinc sulphate combined with traditional treatments, was significantly higher than that in the control group (risk difference = 0.288; 95% CI = 0.087–0.489; p = 0.005; $I^2$ = 94.916%) (Fig 3). However, we observed high heterogeneity among the studies; therefore, we performed a sensitivity test, which confirmed the significant results (Fig 4).

**Subgroup analysis.** In the subgroup analysis, we first categorized patients into two groups based on their initial plasma zinc ion concentration, normal and deficient, using the standard referenced in an article published in 2024 [22]. The results indicated that patients with deficient initial plasma zinc ion levels had significantly better treatment outcomes for viral warts (risk difference = 0.767; 95% CI = 0.649–0.885; p < 0.001) (Fig 5). In contrast, studies involving patients with normal initial plasma zinc ion levels did not show a significant difference in treatment efficacy (risk difference = 0.042; 95% CI = −0.108–0.192; p = 0.579).

We further divided the intervention methods into two groups. One group compared oral zinc sulphate with placebo and labeled monotherapy. The other group combined oral zinc sulphate with cryotherapy, imiquimod, or podophyllin, compared with the use of cryotherapy, imiquimod, or podophyllin alone, and was labeled as combined therapy. The chart shows that in the monotherapy group, the efficacy of zinc sulphate was significantly better than that of the placebo (risk difference = 0.522; 95% CI = 0.350–0.695; p < 0.001) (Fig 6). However, in the combined therapy group, the combination of zinc sulphate with traditional treatments did not show significantly better efficacy than traditional treatments alone (risk difference = 0.017; 95% CI = −0.157–0.192; p = 0.848).

**Meta-regression.** To examine whether the magnitude of the increase in plasma zinc ion levels in the experimental groups after treatment completion and the disease duration were related to treatment efficacy, we conducted a meta-regression analysis. The plasma zinc ion concentration was measured in µg/dL, and the disease duration was measured in months. The results showed that an increase in both zinc ion levels (coefficient = 0.0068; p < 0.001) (Fig 7) and disease duration (coefficient = 0.0233; p = 0.0001) (Fig 8) were significantly associated with treatment efficacy. The greater the increase in zinc ion levels, the more effective the treatment. Unexpectedly, longer disease duration was associated with

**Table 1. Study characteristic of the included trials.**

| Author (Year of Publication) | Country | Blinding | Participants (M/F) Zinc | Participants (M/F) Control | Age Zinc | Age Control | Initial Zinc level (µg/dL) Zinc | Initial Zinc level (µg/dL) Control | Treat/ Follow-up (Month) | Intervention | Outcome measurement |
|---|---|---|---|---|---|---|---|---|---|---|---|
| Gurairi 2002 | Iraq | Single blind [c] | 9/14 | 6/14 | 20 (4-50) | 20 (7-37) | 62.5±10.72 | 66.4 (45.3-85.3) | 2/8 | Zinc: 10mg/kg, max: 600mg/day Control: glucose | Complete disappearance of the lesions without residual scarring |
| Yaghoobi 2009 | Iran | Double blind | 14/18 | 23 | 17.6±5.44 | N/A | 55.09±10.07 | 56.63±8.73 | 2/2 | Zinc: 10mg/kg, max: 600mg/day Control: starch capsule | Disappearance of all warts without residual scarring |
| López-García 2009 | Mexico | Double blind | 25 | 25 | N/A | N/A | 102 [b] | 103 [b] | 2/2 | Zinc: 10mg/kg, max: 600mg/day Control: starch capsule | Total clearance |
| Sadighha 2009 | Iran | Double blind | 5/8 | 13 | 15.9±4.87 | N/A | 53.3±9.7 | 58.04±9.13 | 2/2 | Zinc: 10mg/kg, max: 600mg/day Control: starch capsule | All warts have cleared |
| Akhavan 2014 (Cryotherapy) | Iran | Not mentioned | 0/42 | 0/42 | 20-50 [a] | 20-50 [a] | N/A | N/A | 2/8 | Zinc: 400 mg + cryotherapy Control: cryotherapy | 1. Duration of response to therapy 2. Response to treatment |
| Akhavan 2014 (Imiquimod) | Iran | Not mentioned | 0/42 | 0/42 | 20-50 [a] | 20-50 [a] | N/A | N/A | 2/8 | Zinc: 400 mg + 5% Imiquimod cream three times a week Control: 5% Imiquimod cream three times/ week | 1. Duration of response to therapy 2. Response to treatment |
| Akhavan 2014 (Podophyllin) | Iran | Not mentioned | 0/42 | 0/42 | 20-50 [a] | 20-50 [a] | N/A | N/A | 2/8 | Zinc: 400 mg + 20% Podophyllin solution once a week Control: 20% Podophyllin solution once a week | 1. Duration of response to therapy 2. Response to treatment |
| Moniem 2016 | Egypt | Not mentioned | 7/13 | 8/12 | 34.2±6.98 | 36.2±7.22 | 85.95±25.61 | 81.30±35.87 | 1/1 | Zinc: 10mg/kg, max: 600mg/day Control: no treatment | Lesion improved greater than 75% |
| Mahmoudi 2018 | Iran | Double blind | 24/21 | 18/21 | 45±6.60 | 39±7.47 | 113.77±20.50 | 114.00±21.76 | 2/8 | Zinc: 10mg/kg, max: 600mg/day + cryotherapy every 3 weeks Control: starch capsule + cryotherapy every 3 weeks | Total elimination of all lesions |

Study characteristic of the included trials.

[a] means that the article only mentions the inclusion criteria

[b] only provides mean value, no SD available

[c] patients were blinded

Abbreviation: M = Male; F = Female; N/A = not applicable

Inclusion criteria: 1) randomized controlled trials (RCTs) with human participants; 2) RCTs including the number of individuals who experienced viral wart clearance after treatment in both experimental and control groups; 3) placebo-controlled RCTs (without any age or treatment duration restrictions); and 4) RCTs including those administered a dosage of zinc sulphate, and 5) studies on the oral application of zinc ions.

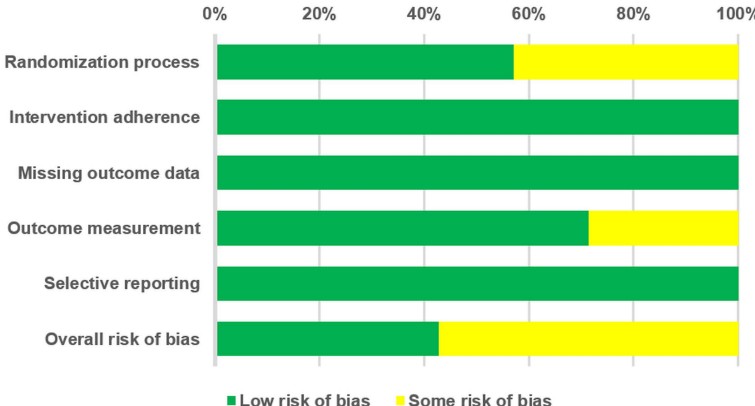

**Fig 2. Summary of quality assessment of studies included in the meta-analysis using Cochrane risk of bias 2 tool.**

**Table 2. Detailed quality assessment of included studies using Cochrane risk of bias 2 tool.**

| First Author | Year | Randomization process | Intervention adherence | Missing outcome data | Outcome measurement | Selective reporting | Overall RoB |
|---|---|---|---|---|---|---|---|
| Gurairi | 2002 | L | L | L | L | L | L |
| Lo´pez-Garcı´a | 2009 | L | L | L | L | L | L |
| Mahmoudi | 2018 | L | L | L | L | L | L |
| Sadighha | 2009 | S[a] | L | L | L | L | S |
| Yaghoobi | 2009 | S[a] | L | L | L | L | S |
| Akhavan | 2014 | L | L | L | S[b] | L | S |
| Moniem | 2016 | S[a] | L | L | S[b] | L | S |

Detailed quality assessment of included studies using Cochrane risk of bias 2 tool

[a]article did not describe the concealment of random allocation.

[b]there might be potential influence of pre-existing knowledge on result interpretation.

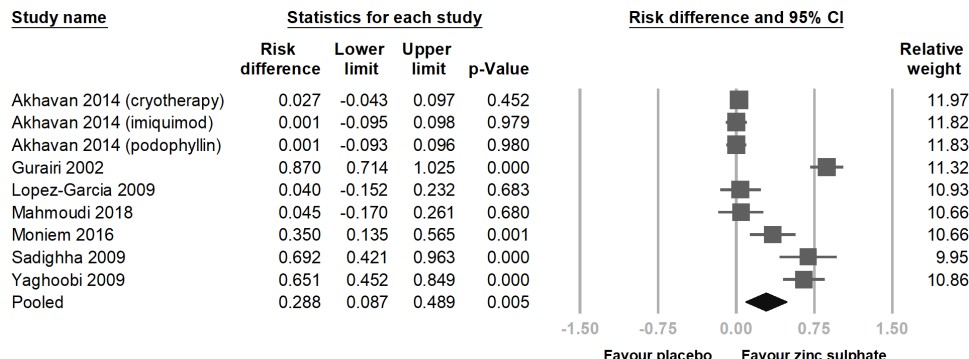

**Fig 3. This is a forest plot comparing the total clearance rate of oral zinc sulphate versus placebo in the treatment of viral warts.** Patients taking oral zinc sulphate had significantly greater total clearance rate of warts. The studies are listed in alphabetical order. CI, confidence interval.

| Study name | Statistics with study removed | | | | Risk difference (95% CI) with study removed |
|---|---|---|---|---|---|
| | Point | Lower limit | Upper limit | p-Value | |
| Akhavan 2014 (cryotherapy) | 0.326 | 0.076 | 0.576 | 0.011 | |
| Akhavan 2014 (imiquimod) | 0.328 | 0.091 | 0.565 | 0.007 | |
| Akhavan 2014 (podophyllin) | 0.328 | 0.090 | 0.566 | 0.007 | |
| Gurairi 2002 | 0.201 | 0.054 | 0.349 | 0.008 | |
| Lopez-Garcia 2009 | 0.319 | 0.099 | 0.539 | 0.004 | |
| Mahmoudi 2018 | 0.318 | 0.099 | 0.536 | 0.004 | |
| Moniem 2016 | 0.281 | 0.065 | 0.497 | 0.011 | |
| Sadighha 2009 | 0.243 | 0.039 | 0.447 | 0.020 | |
| Yaghoobi 2009 | 0.243 | 0.042 | 0.444 | 0.018 | |
| Pooled | 0.288 | 0.087 | 0.489 | 0.005 | |

-1.50  -0.75  0.00  0.75  1.50

Favour placebo    Favour zinc sulphate

**Fig 4. This is the result of sensitivity analysis by using the one-study removal method.** All analyses showed statistically significant effects of oral zinc sulphate in treating viral warts. The studies are listed in alphabetical order. CI, confidence interval.

**Table 3. Summary of the results of the statistical analyses of the included studies.**

| Outcome | | Effect measures[a] | p-Value |
|---|---|---|---|
| Total clearance of viral warts | | RD = 0.288 | p = 0.005 |
| Sensitivity analysis | | RD = 0.288 | p = 0.005 |
| Subgroup analysis | Normal zinc ion level | RD = 0.042 | p = 0.579 |
| | Deficient zinc ion level | RD = 0.767 | p < 0.001 |
| | Monotherapy | RD = 0.522 | p < 0.001 |
| | Combined therapy | RD = 0.017 | p = 0.848 |
| Relapse rates | | log RR = −1.043 | p = 0.001 |
| Treatment-Associated Adverse Event Rates | Nausea | RD = 0.562 | p = 0.020 |
| | Vomiting | RD = 0.205 | p < 0.001 |

RD = risk difference; RR = risk ratio

[a]The zinc sulfate group was set on the positive direction and the control group was set on the negative direction.

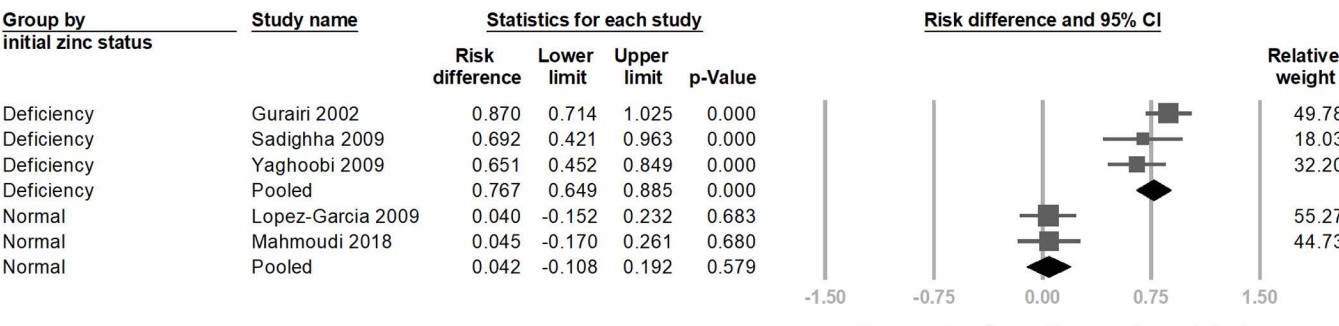

| Group by initial zinc status | Study name | Statistics for each study | | | | Risk difference and 95% CI | Relative weight |
|---|---|---|---|---|---|---|---|
| | | Risk difference | Lower limit | Upper limit | p-Value | | |
| Deficiency | Gurairi 2002 | 0.870 | 0.714 | 1.025 | 0.000 | | 49.78 |
| Deficiency | Sadighha 2009 | 0.692 | 0.421 | 0.963 | 0.000 | | 18.03 |
| Deficiency | Yaghoobi 2009 | 0.651 | 0.452 | 0.849 | 0.000 | | 32.20 |
| Deficiency | Pooled | 0.767 | 0.649 | 0.885 | 0.000 | | |
| Normal | Lopez-Garcia 2009 | 0.040 | -0.152 | 0.232 | 0.683 | | 55.27 |
| Normal | Mahmoudi 2018 | 0.045 | -0.170 | 0.261 | 0.680 | | 44.73 |
| Normal | Pooled | 0.042 | -0.108 | 0.192 | 0.579 | | |

-1.50  -0.75  0.00  0.75  1.50

Favour placebo    Favour zinc sulphate

**Fig 5. This is a forest plot of a subgroup analysis based on patients' initial plasma zinc ion concentration.** The analysis shows that patients with lower initial plasma zinc ion levels have more effective treatment for viral warts when taking zinc sulphate. The studies are listed in alphabetical order. CI, confidence interval.

| Group by combine therapy | Study name | Statistics for each study | | | | Risk difference and 95% CI | Relative weight |
|---|---|---|---|---|---|---|---|
| | | Risk difference | Lower limit | Upper limit | p-Value | | |
| Combined therapy | Akhavan 2014 (cryotherapy) | 0.027 | -0.043 | 0.097 | 0.452 | | 27.35 |
| Combined therapy | Akhavan 2014 (imiquimod) | 0.001 | -0.095 | 0.098 | 0.979 | | 26.31 |
| Combined therapy | Akhavan 2014 (podophyllin) | 0.001 | -0.093 | 0.096 | 0.980 | | 26.41 |
| Combined therapy | Mahmoudi 2018 | 0.045 | -0.170 | 0.261 | 0.680 | | 19.92 |
| Combined therapy | Pooled | 0.017 | -0.157 | 0.192 | 0.848 | | |
| Monotherapy | Gurairi 2002 | 0.870 | 0.714 | 1.025 | 0.000 | | 22.79 |
| Monotherapy | Lopez-Garcia 2009 | 0.040 | -0.152 | 0.232 | 0.683 | | 20.77 |
| Monotherapy | Moniem 2016 | 0.350 | 0.135 | 0.565 | 0.001 | | 19.49 |
| Monotherapy | Sadighha 2009 | 0.692 | 0.421 | 0.963 | 0.000 | | 16.55 |
| Monotherapy | Yaghoobi 2009 | 0.651 | 0.452 | 0.849 | 0.000 | | 20.40 |
| Monotherapy | Pooled | 0.522 | 0.350 | 0.695 | 0.000 | | |

-1.50    -0.75    0.00    0.75    1.50

Favour placebo          Favour zinc sulphate

**Fig 6. This is a forest plot from a subgroup analysis based on whether patients combined traditional therapies.** The analysis indicates that zinc sulphate alone is more effective for treating viral warts compared to combined treatments. However, this might be due to traditional therapies masking the effects of zinc sulphate in the combined treatment group. The studies are listed in alphabetical order. CI, confidence interval.

better treatment outcomes. Summary of the results of the statistical analyses of the included studies are presented in "Tables 3 and 4".

### Secondary outcome

**Relapse rate.** We analyzed the studies that provided data on the number of relapsed cases during the follow-up period. Both studies used zinc sulphate in combination with other traditional treatments. We found that the group using zinc sulphate had a lower probability of relapse (log risk ratio = −1.043; 95% CI = −1.666– −0.420; p = 0.001; $I^2 < 0.1\%$). (Fig 9).

**Treatment-associated adverse event rates.** Among the seven included studies, three provided data on treatment-related adverse effects, encompassing 177 of the 594 participants. Gastrointestinal side effects were the most common, with nausea being the most frequently reported side effect. In all three studies, zinc sulphate use was significantly associated with a higher incidence of nausea (risk difference = 0.562; 95% CI = 0.088–1.036; p = 0.020) (S1 Fig). Vomiting was the second most common side effect and was statistically significant (risk difference = 0.205; 95% CI = 0.092–0.317; p < 0.001) (S2 Fig).

### Reporting biases

We created a funnel plot for the seven studies that visually showed slight asymmetry (S3 Fig). However, Egger's test yielded a two-tailed p value of 0.074, indicating a low likelihood of publication bias. Nonetheless, given the inclusion of fewer than 10 studies, the symmetry should be interpreted with caution.

### Discussion

In this meta-analysis, oral zinc sulphate was effective in achieving total clearance of viral warts, with a statistically significant risk difference (p = 0.005). This significant difference in risk remained consistent in the sensitivity analysis (p = 0.005). In the analysis of relapse, although the number of studies providing such data was limited, those who received traditional therapy combined with oral zinc sulphate experienced significantly fewer wart recurrences within six months of post-treatment than the control group (p = 0.001). In the subgroup analysis, we found that in the monotherapy group, zinc sulphate showed significantly greater efficacy than the placebo; however, in the combined therapy group, the use of zinc sulphate did not result in a significant difference in therapeutic outcomes. Based on the aforementioned findings, zinc sulphate monotherapy may be an effective option for patients who do not wish to undergo conventional

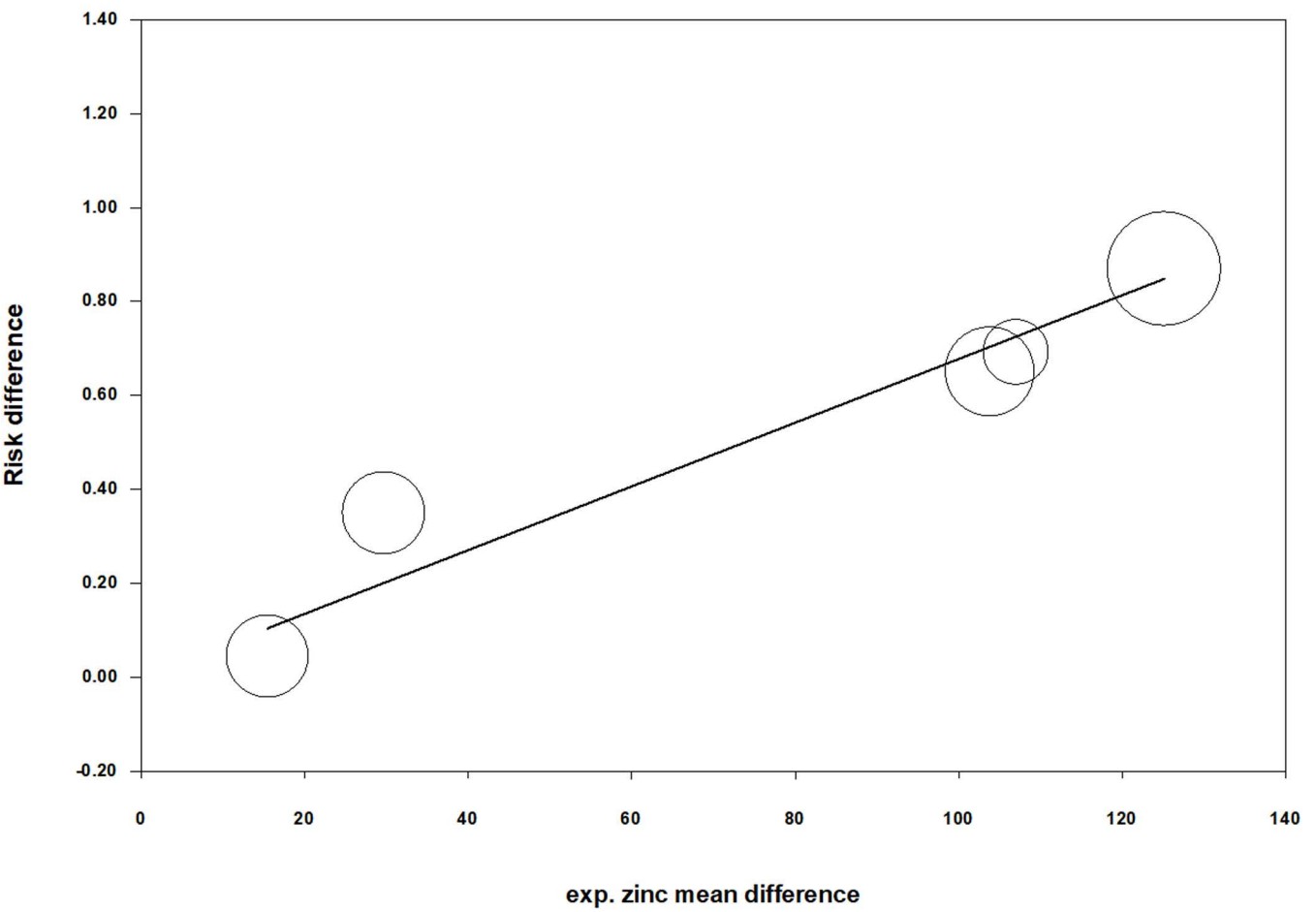

**Fig 7. Meta-regression of risk difference on mean difference of plasma zinc ion concentration (mg/day) in all experimental groups.** The coefficient was 0.0068 with a p value < 0.001.

treatments. In patients receiving traditional therapy for viral warts, the addition of zinc sulphate significantly reduces the recurrence rate. We also found that patients with initially low plasma zinc levels showed a greater treatment response than those with normal zinc levels. Further meta-regression analysis confirmed the relationship between changes in zinc concentration and wart clearance, indicating that larger increases in zinc levels after treatment completion predicted better treatment outcomes. Interestingly, we found that longer disease duration was associated with better treatment efficacy, a point discussed later. Demographic data showed a mean patient age of 31.64 ± 12.49 (SD), aligning with real-world epidemiological findings indicating the highest prevalence among young adults. To the best of our knowledge, this is the first meta-analysis to evaluate the efficacy of oral zinc sulphate for viral warts and to investigate the factors influencing its effectiveness.

A previous prospective study comparing the efficacy of oral zinc sulphate and valacyclovir for wart treatment found that valacyclovir was superior [23]. Another open-label study reported a cure rate of 50% in patients treated with oral zinc sulphate [24]. Two systematic review summarized various zinc-related therapies for wart treatment and highlighted

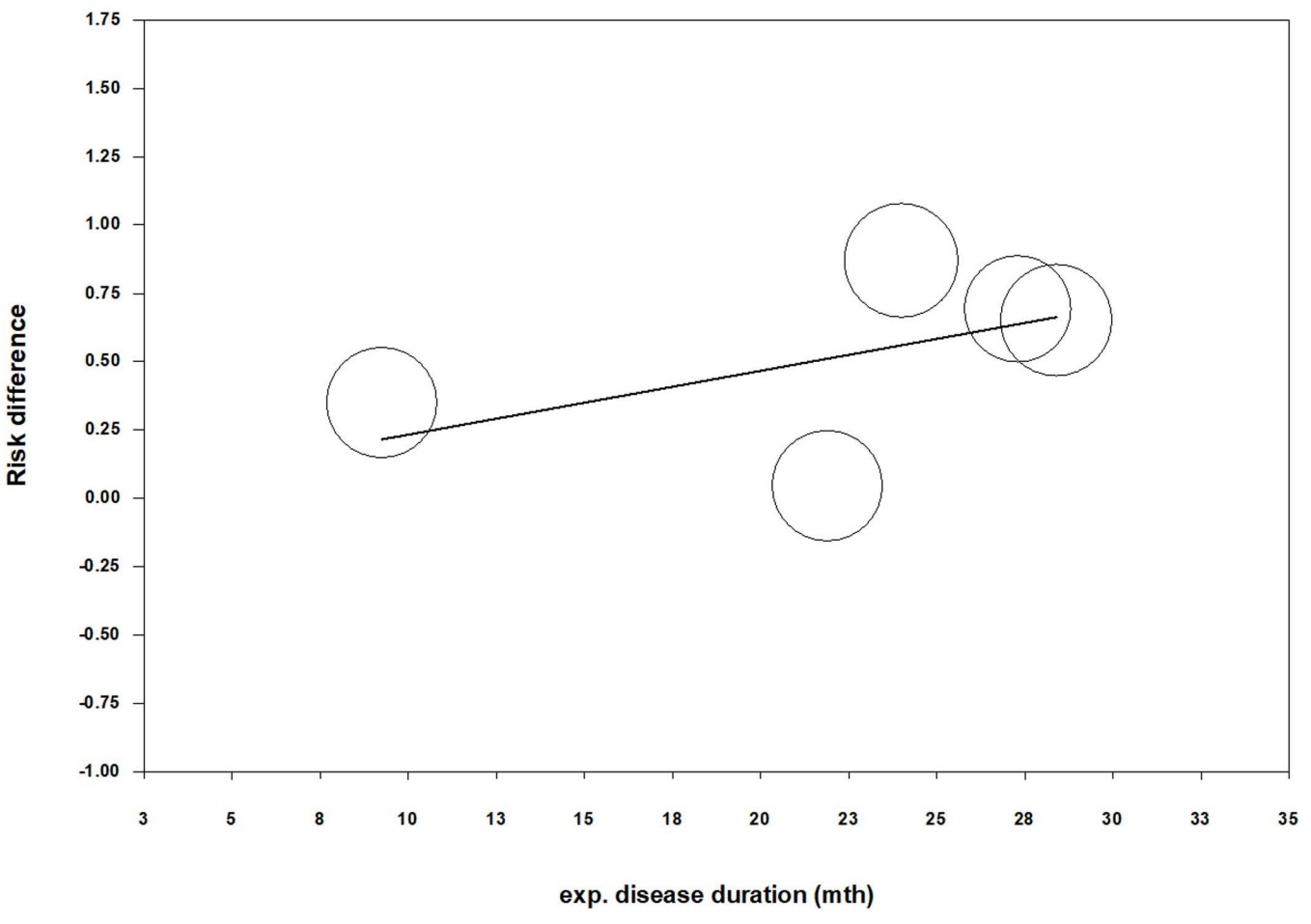

**Regression of Risk difference on exp. disease duration (mth)**

**Fig 8. Meta-regression of risk difference on disease duration (months).** The coefficient was 0.0233 with a p value was 0.0001.

the efficacy of oral zinc sulphate [16,25]. However, the methodologies and experimental designs of the first two studies had limitations. Additionally, the two recently published systematic reviews focused primarily on describing the outcomes and limitations of each clinical trial. While we agree with their summaries regarding the efficacy of oral zinc sulfate, they provided limited analysis of the factors influencing treatment outcomes. This is where our paper adds significant value, as these insights can enhance our understanding of patient selection and the prediction of treatment efficacy.

Some studies suggest that zinc can regulate the translation of interferon-γ through gene methylation or affect them at the post-transcriptional level [26]. Other studies have suggested that zinc promotes and inhibits the signaling pathways of interferon receptors through phosphorylation and dephosphorylation [27–30]. Regarding HPVs, zinc appears to influence E6 and E7 proteins. In high-risk HPV types, exogenous zinc supplementation can effectively inhibit the synthesis of E6 and E7 proteins, leading to apoptosis of dysplastic cervical cancer cells by interrupting the viral life cycle. Additionally, zinc can inhibit the enzymatic activity of viral proteases and polymerases and affect the processes of attachment, infection, and uncoating [31]. Similarly, studies have shown that warts are more refractory to treatment and are more likely to recur in patients with zinc deficiency [6,7,32,33].

**Table 4. Summary of the meta-regression results of the statistical analyses of the included studies.**

| Outcome | Meta-regression by mean difference of increase in zinc ion level post-treatment (µg/dL) | | Meta-regression by mean disease duration (month) | |
|---|---|---|---|---|
| | Coefficient | p-Value | Coefficient | p-Value |
| Total clearance of viral warts | 0.0068 | p<0.001 | 0.0233 | p=0.0001 |

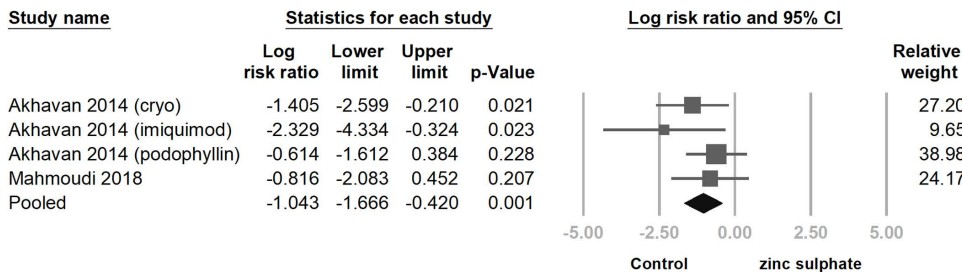

**Fig 9. This is a forest plot comparing the relapse rates within six months for patients treated with oral zinc sulphate combined with traditional therapy versus traditional therapy alone.** The results show that the group receiving zinc sulphate had significantly lower relapse rates. The studies are listed in alphabetical order. CI, confidence interval.

Notably, a considerable proportion of studies examining the relationship between zinc concentration and viral warts, including the articles incorporated in this meta-analysis, originated from Middle Eastern countries. This may be related to the dietary habits in these regions, especially in rural areas where diets often include a high intake of whole wheat bread. This dietary habit is rich in fiber and phytates, which can reduce zinc absorption efficiency, potentially leading to zinc deficiency [34]. The effect of zinc status on wart treatment efficacy was consistently observed in our subgroup analysis. Three randomized controlled trials [10,12,35] included patients with initial plasma zinc concentrations ranging from to 40–60 µg/dL, indicative of mild zinc deficiency [22]. Following oral zinc sulphate treatment, the proportion of patients with total wart clearance was significantly higher than that in the control group (p<0.001). In contrast, two other randomized controlled trials [13,14] involved patients with initial zinc levels above 100 µg/dL, and these studies did not show significant treatment efficacy with zinc sulphate (p=0.579). We further conducted a meta-regression analysis between the mean difference in plasma zinc concentrations before and after treatment and treatment efficacy. A strong positive correlation was found between increased plasma zinc ion concentrations and treatment efficacy (p<0.001).

In the subgroup analysis of treatment methods, using oral zinc sulphate alone for wart treatment resulted in significantly higher total clearance rate compared to the placebo group. Conversely, in the combined therapy group, zinc ions combined with conventional treatment did not significantly outperform conventional treatment alone. We also created an additional analysis excluding studies that used no-treatment controls or glucose as a control, and the results remained consistent (S4 Fig). This finding should be interpreted cautiously, as the trials included in this review show that viral warts respond well to conventional treatments. Given that conventional therapy is already effective, most warts are likely to be resolved, leaving limited room for additional improvement when applying zinc ions. Thus, no synergistic effect appears between zinc and conventional treatment in our study, though zinc may offer a benefit in reducing recurrence rates.

We found that the longer the wart duration, the better the response to zinc sulphate treatment (p=0.0001). This finding contrasts with the current knowledge. In one randomized controlled trial [11], the average disease duration for cases classified as having the best and worst cure rates were 7.65 months and 11.1 months, respectively, with a significant intergroup difference (p=0.013). Another randomized controlled trial from 2018 [13] reported an average disease duration of 16.84 months in the successfully treated group and 24.17 months in the treatment failure group, with a significant

difference (p = 0.03). This unexpected result may be due to the limited number of analyzed studies and scarcity of data on disease duration. We hypothesized that patients with recalcitrant viral warts might be in a state of chronic zinc deficiency, rendering them more resistant to conventional treatment [6,7]. Once zinc sulphate supplementation normalized zinc levels, it appeared that the improved immune status might have contributed to a notable therapeutic effect.

Consumption of zinc sulphate is associated a few side effects. Among the studies included in this meta-analysis, the group receiving zinc sulphate experienced significantly more adverse events, with nausea being the most common and statistically significant (p = 0.02). A 2002 study reported that all patients in the experimental group experienced this side effect [10]. Vomiting was the second most common adverse event, occurring significantly more frequently in the experimental group than in the control group (p < 0.001). In the studies included in this review, the highest dosage was 400–600 mg of zinc sulfate per day, equivalent to an effective zinc ion content of approximately 100–150 mg. In fact, for patients with malabsorption syndromes, such as Crohn's disease or short bowel syndrome, the dose for treating acute zinc deficiency is typically only 50 mg/d [22]. Exceeding this dose can lead to gastrointestinal side effects such as nausea, vomiting, and gastric bleeding [36]. Daily intake of more than 150 mg of zinc can potentially affect lipid profiles and immune function status. Excessive zinc intake may also interfere with copper absorption, leading to copper deficiency [22].

This study had several limitations. First, the number of available studies on this topic was limited, and some were excluded due to their study design. Additionally, some studies had small sample sizes or high dropout rates, which may have contributed to significant heterogeneity and potential bias in the results. To address this, we performed subgroup analyses based on the "initial plasma zinc ion concentration" and "whether combined therapy was used" to examine factors that could contribute to heterogeneity. We also performed a meta-regression to identify the relationship between treatment efficacy and the increase in zinc ion levels as well as the duration of the disease. Secondly, most of the studies included in this meta-analysis originate from the Middle East. Therefore, further research is needed to determine whether this treatment is effective for patients in other regions. While conducting subgroup analyses by ethnicity could be a viable approach to explore differences in treatment efficacy, in this study, the limited number of available articles led to a reduction in power for certain subgroups. As a result, we did not include ethnicity in our subgroup analysis. Third, the sex distribution of the patients included in this study was imbalanced, with only 27.4% of the patients being male. However, whether zinc sulphate is particularly effective in female remains unclear. Most of the existing literature has not analyzed sex as a subgroup. Future studies should address this issue.

## Conclusion

Oral zinc sulphate monotherapy can provide significant therapeutic effects in patients unwilling to undergo conventional treatment. For patients using traditional treatments for viral warts, the addition of zinc sulphate can significantly reduce the recurrence rate over the next six months. Future research should focus on the safety of zinc sulphate at higher concentrations, considering cases from other regions or ethnic groups, and explore the relationship between the disease duration and its impact on the treatment efficacy for viral warts.

## Supporting information

**S1 Fig. This is a forest plot of the risk difference of nausea when taking oral zinc sulphate.** Nausea was significantly more frequent in patients taking zinc sulphate. The studies are listed in alphabetical order. CI, confidence interval. (TIF)

**S2 Fig. This is a forest plot of the risk difference of vomiting when taking oral zinc sulphate.** Vomiting was significantly more frequent in patients taking zinc sulphate. The studies are listed in alphabetical order. CI, confidence interval. (TIF)

**S3 Fig. The funnel plot of included trials.** The p value of the Egger's test was 0.074, indicating probably no evidence of publication bias.
(TIF)

**S4 Fig. This is a forest plot of subgroup analysis conducted based on treatment methods, excluding studies using no-treatment control and glucose as control.** After reanalysis, the results were consistent with the original subgroup analysis.
(TIF)

**S1 Table. PRISMA 2020 checklist.**
(DOCX)

**S2 Table. Searching process.**
(DOCX)

**S3 Table. All studies identified.**
(XLSX)

**S4 Table. All data extracted.**
(DOCX)

**S1 Data. Minimal dataset.**
(XLSX)

## Author contributions

**Conceptualization:** Chen-Chi Wang, Wei-Xiang Wang.

**Methodology:** Chen-Chi Wang, Wei-Xiang Wang.

**Supervision:** Po-Yuan Wu.

**Writing – original draft:** Chen-Chi Wang, Wei-Xiang Wang, Po-Yuan Wu.

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
