## [Decision Letter · Decision Letter 0]

23 Oct 2024

PONE-D-24-34354Oral zinc sulphate reduces recurrence rate and provides significant therapeutic effects for viral warts: a systematic review and meta-analysis of randomized controlled trialsPLOS ONE

Dear Dr. Wu,

Thank you for submitting your manuscript to PLOS ONE. After careful consideration, we feel that it has merit but does not fully meet PLOS ONE’s publication criteria as it currently stands. Therefore, we invite you to submit a revised version of the manuscript that addresses the points raised during the review process.

We look forward to receiving your revised manuscript.

Kind regards,

Vineet Kumar Rai, PhD

Academic Editor

PLOS ONE

Journal Requirements: When submitting your revision, we need you to address these additional requirements. 1. Please ensure that your manuscript meets PLOS ONE's style requirements, including those for file naming. The PLOS ONE style templates can be found at https://journals.plos.org/plosone/s/file?id=wjVg/PLOSOne_formatting_sample_main_body.pdf and https://journals.plos.org/plosone/s/file?id=ba62/PLOSOne_formatting_sample_title_authors_affiliations.pdf 2. As required by our policy on Data Availability, please ensure your manuscript or supplementary information includes the following:  A numbered table of all studies identified in the literature search, including those that were excluded from the analyses.   For every excluded study, the table should list the reason(s) for exclusion.   If any of the included studies are unpublished, include a link (URL) to the primary source or detailed information about how the content can be accessed.  A table of all data extracted from the primary research sources for the systematic review and/or meta-analysis. The table must include the following information for each study:  Name of data extractors and date of data extraction  Confirmation that the study was eligible to be included in the review.   All data extracted from each study for the reported systematic review and/or meta-analysis that would be needed to replicate your analyses.  If data or supporting information were obtained from another source (e.g. correspondence with the author of the original research article), please provide the source of data and dates on which the data/information were obtained by your research group.  If applicable for your analysis, a table showing the completed risk of bias and quality/certainty assessments for each study or outcome.  Please ensure this is provided for each domain or parameter assessed. For example, if you used the Cochrane risk-of-bias tool for randomized trials, provide answers to each of the signalling questions for each study. If you used GRADE to assess certainty of evidence, provide judgements about each of the quality of evidence factor. This should be provided for each outcome.   An explanation of how missing data were handled.  This information can be included in the main text, supplementary information, or relevant data repository. Please note that providing these underlying data is a requirement for publication in this journal, and if these data are not provided your manuscript might be rejected. 3. We notice that your supplementary figures are uploaded with the file type 'Figure'. Please amend the file type to 'Supporting Information'. Please ensure that each Supporting Information file has a legend listed in the manuscript after the references list.

Reviewers' comments:

Reviewer's Responses to Questions

**Comments to the Author**

1. Is the manuscript technically sound, and do the data support the conclusions?

Reviewer #1: Partly

Reviewer #2: Yes

Reviewer #3: Yes

Reviewer #4: Yes

Reviewer #5: Partly

2. Has the statistical analysis been performed appropriately and rigorously? 

Reviewer #1: I Don't Know

Reviewer #2: Yes

Reviewer #3: Yes

Reviewer #4: I Don't Know

Reviewer #5: Yes

3. Have the authors made all data underlying the findings in their manuscript fully available?

Reviewer #1: No

Reviewer #2: Yes

Reviewer #3: Yes

Reviewer #4: No

Reviewer #5: Yes

4. Is the manuscript presented in an intelligible fashion and written in standard English?

Reviewer #1: No

Reviewer #2: Yes

Reviewer #3: Yes

Reviewer #4: Yes

Reviewer #5: Yes

5. Review Comments to the Author

Reviewer #1: Thank you for inviting me to review the paper on the effect of oral zinc sulfate on viral warts. My main concern is why the authors pooled the Akhavan study, which has a different drug dosage, with other studies.

Other points:

Title:

Reduces "the" recurrence rate...

Abstract:

The abstract needs to be structured. The search date and the databases searched should be mentioned (method).

The number of included studies and the number of studies in your primary search should be mentioned (result).

Method:

Lines 96-98: Did the authors use MeSH words and other synonyms?

Was any language restriction applied?

Result:

Summarize the number of studies at each step of the selection of eligible studies in the study selection part.

A funnel plot should not be used for studies with fewer than 10 inclusions.

Add the list of excluded studies at the full-text review step and the reasons for exclusion in the supplementary materials.

A table should be added summarizing the results of the statistical analyses of the included studies, such as RR, RD, etc.

Data regarding double/single blinding of the studies should be added to Table 1.

The inclusion criteria of the RCTs should be added to the table.

Discussion:

Why were these studies only conducted in the Middle East and North Africa region?

Reviewer #2: Strengths: This systematic review and meta-analysis offers robust and comprehensive insights into the therapeutic effects of oral zinc sulphate for viral warts. By conducting subgroup analyses based on plasma zinc levels and treatment modalities, the study provides more detailed insights into the efficacy of zinc sulphate under various conditions. The use of statistical methods such as random-effects models, sensitivity analyses, and Egger’s test for publication bias ensures that the conclusions are not only statistically significant but also reliable. Clinically relevant information is particularly highlighted for patients with low plasma zinc levels or those resistant to conventional treatments.

Limitations: Some studies included in the meta-analysis had small sample sizes, which may reduce the statistical power of the conclusions. Additionally, the study population was predominantly female (72.6%), and the impact of zinc supplementation may differ across sexes.

Reviewer #3: The authors conducted an interesting study on zinc and viral wart.

It would be better if you could provide evidence for the statement "However, the conclusions of these studies are inconsistent, and comprehensive research has been lacking in recent years.".

Reviewer #4: 1. English writing is generally not indented at the beginning of the line.

2. L67-68, please add to the preamble the inconsistent specifics of the oral zinc findings and add references, which is what is needed to do this study.

3. there are ethnographic differences in the incidence of warts, and although the literature is limited, the reason why this was not subgrouped needs to be stated in the limitations

4. please provide a specific search strategy for each database

5. the search strategy mentions references to included literature as sources for inclusion, this needs to be reflected in the flowchart

6. Please add the specific number of literature screening entries in the text.

7. The small amount of literature included is still a major concern, especially when the amount of literature is small for regression.

8. Two systematic evaluations have been conducted, and the strengths of this study are not fully described in the discussion.

9. Why were subgroup analyses of gender not conducted, and if the gender differences were so large, was it possible to extract data for secondary analysis?

Reviewer #5: Comments to PLOS ONE PONE-D-24-34354

Title:

Oral zinc sulphate reduces recurrence rate and provide significant therapeutic effects for viral warts: a systematic review and meta-analysis of randomized controlled trials

Comments to authors

Thank you for your hard work. I appreciate the efforts of the authors.

• Through a systematic review and meta-analysis, this study has reported an evidence on the efficacy of oral zinc sulphate, whether monotherapy or as a combined with traditional therapies compared to placebo or other oral therapies for the treatment of viral warts.

Recommendation for improvement:

• The study provided new insights, however there are several weaknesses still need to be considered.

• Deciding to conduct a systematic review, meta-analysis, and meta-regression to produce new, more accurate and valid evidence, however, the interpretations of the findings are not clearly presented. Therefore, the authors are asked to discuss with biostatistician or epidemiologist to enriched the presentation of findings.

• The writing style needs to be considered to make it easier for readers to understand and to implement the findings of this study.

Introduction

• The background that has become the basis for this research is not strong enough to describe the existing problems. What are the research questions that the authors want to answer? Are all parts (population, intervention, comparison, outcome) that have become the target of the authors supported by data, for example from results of previous research or from existing theories are to be tested? The authors are asked to consider on how is this background related to the research conducted that can help strengthen the authors’ hypotheses. Therefore, the authors are advised to explain the relevance of the background presented.

• Line 48: The authors stated, “… with the highest prevalence observed in school-aged children and young adults”. Are the results of your study in line with this statement? The results you have reported are the mean (+SD) age of 31.64+12.49. If it is in accordance with the results of previous research that is us as a reference, then it needs to be emphasized in the discussion section.

• Line 66: “According to the current literature, both topical and oral zinc formulations may be effective as primary or adjunct therapies for viral warts”. However, this systematic review and meta-analysis only analyzed the effectiveness of oral administration. The authors are asked to explain the reason why this study only focused on oral administration.

• In this background section, it would be better to also explain the results of previous primary studies that describe of how effective the zinc sulphate can be as monotherapy or combined therapy, or how the intervention might work compared with others.

Methods

• Line 80: “P, human participants with skin or external genital warts”. The authors are asked to provide scientific reasons why only skin warts and external genital warts are included in the criteria, while all types of warts need to be treated.

• Line 90: The authors stated “the exclusion criteria were..1) non-RCTs; 2) RCTs not involving oral zinc sulphate supplementation , 4) non-placebo-controlled,,,” Explanation of the things that were exclusion criteria are actually already described as inclusion criteria. For example, in the inclusion criteria it is clearly written that studies with RCTs design will be included. This means (without needing to be written again) that studies with designs other than RCTs will be excluded, so it may not need to be sharpened again by writing it in the exclusion criteria section.

• Line 95: In the literature search strategy, the authors only limited the search by using 5 electronic databases. The authors should explain this, as well as explain whether they used grey literature or hand-searching to recruit includes study in this systematic review.

• Line 96: The keywords compiled by the authors seem to describe only intervention and outcome. Therefore, authors are asked to consider developing a search strategy by enriching the keywords that include participants and comparisons as well. Boolean technique and the use of more alternative words need to be considered.

• The authors are asked to explain if there is any language restriction, and the reasons to limit it.

• Line 102: “If the two authors had different opinions during the process, a third author was conducted to resolve any disagreements”. There, the authors are asked to explain if there is any disagreement, what are them, and how to solve them.

• Line 116: The secondary outcome was defined as the rate of relapse occurring within six months after treatment as well as any treatment-related side effects. Please be explained clearly if you analyses only for studies reported rate of relapse at least 6 months after intervention.

• Line 112, to consistent with the results especially for randomization (in line 185: “..three studies were classified as having some risk of bias in the randomization process because they did not describe the concealment of random allocation”), the authors are suggested to define clearly about the randomization.

Results

• Line 153: “Articles without full-text availability were excluded”. Please be explained if any effort in finding out this article, or how the authors solve this issue.

• Line 153: “… and we ensured that all zinc sulphate studies were administered orally..”. This statement should be in line with inclusion criteria

• Line 161: “27.4% of whom were male”. In this part, why are male highlighted, usually what is highlighted is the dominant characteristic, in this case are women. Unless there is a specific reason of male participation.

• Line 162: “Based on the available data, the average age was 31.64+12.49” It would be better if these findings are explained briefly in the discussion section, if necessary, because in the introduction section, the authors pay attention to the age variable so that it will be seen whether your results support the explanation in the introduction.

• Table 1: The authors stated that data from the Akhavan, 2014 study (either Cryotherapy or Imiquimod) the comparator was a cream, not an oral preparation. It would be better if this be explained in the methods or inclusion criteria that the comparator is not limited to oral preparations only.

• Line 261: “Unexpectedly, longer disease duration was associated with better treatment outcomes”. The authors are mentioned this issue in the Discussion as disagreement with only one RCT. Therefore, the authors are asked to suggest in Conclusion section what kind of research could be done to strengthen this finding.

• Figure 2: Process of randomization was less than 60%, and overall low risk of bias of included studies was less than 50%. The authors are asked to explain about the risk of bias of included studies, in line with decided to do meta-regression. That may affect the methodological quality of the study.

• Figure 6: the authors are asked to explain on the reasons to combine the combined therapy and monotherapy in one.

• Supplement figure 4: the authors are asked to explain the results, where 5 out of 8 studies are outside the line.

• The high heterogeneity may also be caused by inconsistent of operational definition of variables, design, or dosage of zinc sulphate, and duration for therapies.

Discussion:

• The Discussion section are provided insight findings. However, the authors are lacking in explanation of the agreement and disagreement of this findings with previous studies or literature theories. The authors are asked to clearly distinguish finding gap in the background and how your results contribute to the field.

• The authors are suggested to enriched their arguments, incorporate multiple viewpoints from existing literature to provide a balanced understanding of the topic.

• The authors are also advised to discuss deeply about the potential of the limitations of the study in the end of Discussion section.

Conclusions

• The authors are suggested to explain any unique contributions, potential applications, or scientific and policy implementation of the findings.

6. PLOS authors have the option to publish the peer review history of their article (what does this mean? ). If published, this will include your full peer review and any attached files.

**Do you want your identity to be public for this peer review?** For information about this choice, including consent withdrawal, please see our Privacy Policy .

Reviewer #1: No

Reviewer #2: No

Reviewer #3: No

Reviewer #4: No

Reviewer #5: **Yes: ** Windy Mariane Virenia Wariki

---

## [Author Response · Author response to Decision Letter 1]

10 Dec 2024

Reviewer #1

Q1: Thank you for inviting me to review the paper on the effect of oral zinc sulfate on viral warts. My main concern is why the authors pooled the Akhavan study, which has a different drug dosage, with other studies.

A1: Thank you very much for taking the time to read my research. It is truly an honor to receive your valuable feedback. In the Akhavan study, the zinc sulfate dosage was fixed at 400 mg/day, which indeed differs from other studies (10 mg/kg/day, with a maximum of 600 mg/day). Currently, there is no literature specifying the exact zinc sulfate dosage for treating warts. However, when referencing studies on other diseases, such as Sharquie's publication (1) on “oral zinc sulfate treatment for leishmaniasis,” it was mentioned that there was no significant difference in treatment efficacy between the 5 mg/kg and 10 mg/kg groups. For an adult weighing 60 kg, the daily intake would be between 300 and 600 mg. Therefore, we believe that keeping the zinc sulfate dosage within this range should be considered a reasonable experimental design, and this is the reason we pooled Akhavan study together with others. We sincerely appreciate your insights.

Q2: Title: Reduces "the" recurrence rate...

A2: Thank you for pointing out this significant grammatical error. It has been corrected!

Q3: The abstract needs to be structured. The search date and the databases searched should be mentioned (method). The number of included studies and the number of studies in your primary search should be mentioned (result).

A3: Thank you very much for your valuable feedback. We have incorporated the essential elements you mentioned into the abstract. Below is the revised paragraph. [On July 1, 2024, we performed an extensive database search on PubMed, Embase, Web of Science, Cochrane CENTRAL, and ClinicalTrials.gov. Initially, 952 studies were identified, and after screening, 7 studies were included in the final analysis.]

Q4: Did the authors use MeSH words and other synonyms? Was any language restriction applied?

A4: Thank you for raising this important question. Although we did not use MeSH terms in our search, we made every effort to include relevant keywords and their synonyms in the search strategy. For example, we accounted for the various terminologies used for warts in different forms and considered the potential formulations of zinc-containing medications. Additionally, we did not impose any language restrictions during the literature search.

Q5: Summarize the number of studies at each step of the selection of eligible studies in the study selection part. Add the list of excluded studies at the full-text review step and the reasons for exclusion in the supplementary materials.

A5: Thank you for your feedback. We have updated the content in the manuscript accordingly! The following is the revised section. [Initially, we identified 952 studies. We then removed duplicates (n = 298), studies on unrelated topics (n = 554), and non-RCT studies (n = 60), leaving 40 studies. Next, we excluded studies without full-text availability (n = 2), those not administering zinc ions orally (n = 26), and studies with inappropriate experimental or control group designs (n = 5). This resulted in a final selection of 7 studies for analysis. The literature searching process is presented in “Fig 1”, and all non-duplicate studies (n = 654) identified during the search, along with the reasons for their exclusion, are documented in “S3_Table”.]

Q6: A funnel plot should not be used for studies with fewer than 10 inclusions.

A6: Indeed, you are absolutely correct. Conducting a funnel plot with fewer than 10 studies may lead to misinterpretations regarding potential asymmetry. Our main reason for producing the funnel plot is to transparently present the data characteristics, allowing readers to intuitively understand the relationship between the distribution of studies and research findings. Although we have fewer than 10 studies, we also applied Egger's test to check for asymmetry, and the results did not indicate any. Therefore, we interpret the broader horizontal scatter as a reflection of heterogeneity among studies. A possible improvement would be to clearly mention in the Results section that this funnel plot has limitations due to the small number of studies analyzed, which should be interpreted cautiously. [We created a funnel plot for the seven studies that visually showed slight asymmetry. However, Egger's test yielded a two-tailed p value of 0.074, indicating a low likelihood of publication bias. Nonetheless, given the inclusion of fewer than 10 studies, the symmetry should be interpreted with caution.]

Q7: A table should be added summarizing the results of the statistical analyses of the included studies, such as RR, RD, etc.

A7: Thank you very much for this valuable feedback; it will make our research findings more accessible to readers. We have now added table 3 and 4 into the paper.

Q8: Data regarding double/single blinding of the studies should be added to Table 1. The inclusion criteria of the RCTs should be added to the table.

A8: Thank you very much for this important feedback; it has made the study characteristics more comprehensive. The updated table has been included in the manuscript.

Q9: Why were these studies only conducted in the Middle East and North Africa region?

A9: Indeed, you point out an important issue. The majority of the studies included in this paper are from the Middle East and North Africa. We speculate that this may be related to healthcare accessibility in these regions, which might make viral warts a significant concern in these areas. On the other hand, from the data we reviewed, although the prevalence rate of warts among Whites is reported to be twice as high as that among Blacks and Asians, there are also isolated reports indicating a wart prevalence of 3.3% in the United States and as high as 33% in the Netherlands (2, 3). Clearly, there is a notable discrepancy between regional and overall prevalence rates. As there are currently no prevalence reports from the Middle East, we hypothesize that the prevalence of warts may also be relatively high in the Middle East and North Africa, making this a frequent concern for dermatologists in these areas, which may explain the higher number of studies on this topic originating from these regions.

Reviewer #2

Q1: Strengths: This systematic review and meta-analysis offers robust and comprehensive insights into the therapeutic effects of oral zinc sulphate for viral warts. By conducting subgroup analyses based on plasma zinc levels and treatment modalities, the study provides more detailed insights into the efficacy of zinc sulphate under various conditions. The use of statistical methods such as random-effects models, sensitivity analyses, and Egger’s test for publication bias ensures that the conclusions are not only statistically significant but also reliable. Clinically relevant information is particularly highlighted for patients with low plasma zinc levels or those resistant to conventional treatments.

A1: Thank you very much for taking the time to carefully read our article. We greatly appreciate your detailed insights and thorough understanding of the important aspects of this study. The results presented in this article focus on the effectiveness of oral zinc sulfate in treating viral warts. In addition to reporting the overall outcomes, our study distinguishes itself from other systematic reviews by utilizing subgroup analysis and meta-regression to identify potential factors that may influence treatment efficacy. These factors are then integrated with clinical practice to facilitate shared decision-making between physicians and patients who may consider this therapy.

Q2: Limitations: Some studies included in the meta-analysis had small sample sizes, which may reduce the statistical power of the conclusions. Additionally, the study population was predominantly female (72.6%), and the impact of zinc supplementation may differ across sexes.

A2: Indeed, you are correct. One of the weaknesses of this study lies in the smaller sample sizes of certain included studies, as smaller studies may be more prone to showing significant effects. However, we addressed this issue by conducting a sensitivity test, which helps us understand whether the primary outcome remains significant after the sequential removal of individual studies. Regarding the gender distribution, the included studies tended to have a higher proportion of female participants. Unfortunately, we were unable to obtain data on treatment outcomes stratified by gender from the included studies. As a result, we could not perform an additional meta-analysis to evaluate whether the effectiveness of oral zinc sulfate differs between genders. Thank you for your valuable feedback.

Reviewer #3

Q1: It would be better if you could provide evidence for the statement "However, the conclusions of these studies are inconsistent, and comprehensive research has been lacking in recent years."

A1: Yes, your question is very important, and it has also made us realize that our description of the research background in the article was not sufficiently precise. To strengthen the discussion on the inconsistent evidence in existing literature, here is the revised paragraph. [Existing literature on oral zinc therapy for viral warts shows mixed findings: one clinical trial reported excellent therapeutic effects, with 86% of patients taking zinc supplements achieving complete wart clearance, while none were cured in the placebo group (4). Other studies suggest mild to moderate benefits (5, 6), while some oppose these findings, stating that zinc supplements neither cure viral warts nor provide additional benefits (7-9). Since the conclusions of these studies are inconsistent and comprehensive research has been lacking…]

Reviewer #4

Q1: English writing is generally not indented at the beginning of the line.

A1: Understood, I have removed the indentation at the beginning of the lines. Thank you for your guidance!

Q2: Please add to the preamble the inconsistent specifics of the oral zinc findings and add references, which is what is needed to do this study.

A2: Indeed, we did not provide a detailed description of our study background and the inconsistent outcomes in the existing literature. Allow me to elaborate; here is the revised paragraph. [Existing literature on oral zinc therapy for viral warts shows mixed findings: one clinical trial reported excellent therapeutic effects, with 86% of patients taking zinc supplements achieving complete wart clearance, while none were cured in the placebo group (4). Other studies suggest mild to moderate benefits (5, 6), while some oppose these findings, stating that zinc supplements neither cure viral warts nor provide additional benefits (7-9). Since the conclusions of these studies are inconsistent and comprehensive research has been lacking…]

Q3: There are ethnographic differences in the incidence of warts, and although the literature is limited, the reason why this was not subgrouped needs to be stated in the limitations.

A3: Indeed, incorporating ethnicity as a subgroup in our analysis to explore differences in treatment efficacy is a feasible approach. However, the number of studies available on this topic is relatively limited. This limitation would result in certain ethnic subgroups containing only a single study, which significantly reduces the power of these subgroups. We have added these reasons to the limitations section. Below is the revised paragraph. [Secondly, most of the studies included in this meta-analysis originate from the Middle East. Therefore, further research is needed to determine whether this treatment is effective for patients in other regions. While conducting subgroup analyses by ethnicity could be a viable approach to explore differences in treatment efficacy, in this study, the limited number of available articles led to a reduction in power for certain subgroups. As a result, we did not include ethnicity in our subgroup analysis.]

Q4: Please provide a specific search strategy for each database

A4: We sincerely apologize for not including the document initially. We have now added the searching strategy used for each database to S2 Table.

Q5: the search strategy mentions references to included literature as sources for inclusion, this needs to be reflected in the flowchart

A5: Thank you for your attention to detail; we truly appreciate the time you've taken to review our figures. We acknowledge that we initially overlooked this detail, which was indeed an oversight on our part. We have now added the cross-reference item to Fig 1.

Q6: Please add the specific number of literature screening entries in the text.

A6: Thank you for pointing this out; we appreciate your attention to detail, as this was indeed an oversight on our part. Below is the revised paragraph. [Initially, we identified 952 studies. We then removed duplicates (n = 298), studies on unrelated topics (n = 554), and non-RCT studies (n = 60), leaving 40 studies. Next, we excluded studies without full-text availability (n = 2), those not administering zinc ions orally (n = 26), and studies with inappropriate experimental or control group designs (n = 5). This resulted in a final selection of 7 studies for analysis.]

Q7: The small amount of literature included is still a major concern, especially when the amount of literature is small for regression.

A7: Indeed, as noted in the first point of the limitations section, this meta-analysis has limitations in terms of the number of studies and the sample sizes of patients included in each study. I fully agree with your insight on this matter. We have made every effort to search for all potentially relevant literature, and we hope that future RCTs from various regions will further investigate this topic. You also mentioned the issue of regression analysis. It is true that conducting meta-regression with a limited number of studies presents challenges and risks; however, we have provided p-values for each meta-regression performed. While the statistical power may not be strong, we believe these findings still hold some reference value.

Q8: Two systematic evaluations have been conducted, and the strengths of this study are not fully described in the discussion.

A8: Indeed, you've clearly identified a significant weakness in the article. To clarify the differences between this paper and other existing studies for our readers, we have revised the discussion section. Here is the modified paragraph. [Additionally, the two recently published systematic reviews focused primarily on describing the outcomes and limitations of each clinical trial. While we agree with their summaries regarding the efficacy of oral zinc sulfate, they provided limited analysis of the factors influencing treatment outcomes. This is where our paper adds significant value, as these insights can enhance our understanding of patient selection and the prediction of treatment efficacy.]

Q9: Why were subgroup analyses of gender not conducted, and if the gender differences were so large, was it possible to extract data for secondary analysis?

A9: Indeed, your point is very precise. In our included population, the proportion of male participants was significantly lower than female participants. Using gender as a basis for subgroup analysis might reveal differences in the effects of zinc sulphate across genders. However, while all studies provided the number of participants experiencing relief in both the intervention and control groups, they did not specify these figures for males and females separately. This lack of data prevented us from conducting a subgroup analysis by gender. We have addressed this issue in the third part of the limitations section and hope that future studies will consider documenting treatment effects by gender.

Reviewer #5

Q1: Through a systematic review and meta-analysis, this study has reported an evidence on the efficacy of oral zinc sulphate, whether monotherapy or as a combined with traditional therapies compared to placebo or other oral therapies for the treatment of viral warts.

Recommendation for improvement:

•

---

## [Decision Letter · Decision Letter 1]

1 Apr 2025

Oral zinc sulphate reduces the recurrence rate and provides significant therapeutic effects for viral warts: a systematic review and meta-analysis of randomized controlled trials

PONE-D-24-34354R1

Dear Dr.Po-Yan,

We’re pleased to inform you that your manuscript has been judged scientifically suitable for publication and will be formally accepted for publication once it meets all outstanding technical requirements.

Kind regards,

Laith Naser Al-Eitan, Ph.D

Academic Editor

PLOS ONE

Additional Editor Comments (optional):

Reviewers' comments:

Reviewer's Responses to Questions

**Comments to the Author**

1. If the authors have adequately addressed your comments raised in a previous round of review and you feel that this manuscript is now acceptable for publication, you may indicate that here to bypass the “Comments to the Author” section, enter your conflict of interest statement in the “Confidential to Editor” section, and submit your "Accept" recommendation.

Reviewer #1: All comments have been addressed

Reviewer #4: All comments have been addressed

Reviewer #5: All comments have been addressed

2. Is the manuscript technically sound, and do the data support the conclusions?

Reviewer #1: (No Response)

Reviewer #4: Yes

Reviewer #5: Yes

3. Has the statistical analysis been performed appropriately and rigorously? 

Reviewer #1: (No Response)

Reviewer #4: Yes

Reviewer #5: Yes

4. Have the authors made all data underlying the findings in their manuscript fully available?

Reviewer #1: (No Response)

Reviewer #4: Yes

Reviewer #5: Yes

5. Is the manuscript presented in an intelligible fashion and written in standard English?

Reviewer #1: (No Response)

Reviewer #4: Yes

Reviewer #5: Yes

6. Review Comments to the Author

Reviewer #1: (No Response)

Reviewer #4: The authors have made appropriate revisions based on the reviewers' suggestions and all issues have been addressed without further suggestions.

Reviewer #5: I would like to express my heartfelt gratitude for the authors' hard work and dedication in revising the manuscript on the basis of the feedback provided. The authors' commitment to improving the quality of the manuscript is truly commendable. The revisions made reflect a deep understanding of the subject matter and show the willingness to engage with constructive criticism.

7. PLOS authors have the option to publish the peer review history of their article (what does this mean? ). If published, this will include your full peer review and any attached files.

**Do you want your identity to be public for this peer review?** For information about this choice, including consent withdrawal, please see our Privacy Policy .

Reviewer #1: No

Reviewer #4: No

Reviewer #5: No

---

## [Editor Report · Acceptance letter]

PONE-D-24-34354R1

PLOS ONE

Dear Dr. Wu,

I'm pleased to inform you that your manuscript has been deemed suitable for publication in PLOS ONE. Congratulations! Your manuscript is now being handed over to our production team.

Kind regards,

on behalf of

Professor Laith Naser Al-Eitan

Academic Editor

PLOS ONE